# Cognitive flexibility and sociality in Guinea baboons (Papio papio)

**Julie Gullstrand**[1,2,3¤]*, **Nicolas Claidière**[1,2,3¤], **Joël Fagot**[1,2,3¤]

**1** Centre de Recherche en Psychologie et Neurosciences, CNRS UMR 7077, Aix-Marseille Université, Marseille, France, **2** Station de Primatologie-Celphedia, CNRS UAR 846, Rousset, France, **3** Institute for Language, Communication and the Brain, Aix-Marseille Université, CNRS, Aix-en-Provence, France

¤ Current address: Centre de Recherche en Psychologie et Neurosciences, Aix-Marseille Université, Marseille, France

* Julie.gullstrand@univ-amu.fr

**Data Availability Statement:** All files are available from OSF database at: DOI 10.17605/OSF.IO/JNW82.

**Funding:** The author(s) received no specific funding for this work.

## Abstract

Cognitive flexibility is an executive function playing an important role in problem solving and the adaptation to contextual changes. While most studies investigated the contribution of cognitive flexibility to solve problems in the physical domain, the current study on baboons (*Papio papio*) investigated its contribution to sociality. The current study verified whether there is a relationship between cognitive flexibility at the individual level and the position of the individuals within their social group. Our study re-analysed for that purpose an already published dataset of 18 baboons Guinea baboons tested over two years in an adaptation of the Wisconsin Card Sorting task. The dominance rank and social network were inferred from their free access to the computer test system on which the cognitive task was presented. We found no clear-cut relationship between the hierarchical rank and cognitive flexibility (perseveration, learning latency and response time). By contrast, the most central baboons in their social network are those with the best performance in terms of cognitive flexibility. Overall, this study confirms our hypothesis that cognitive flexibility plays some roles in the regulation of the social relationship in baboons.

## Introduction

Non-human primates must cope with frequent environmental changes. In the wild, their access to water or food resources often varies with the seasons, and the type of food resources, the distribution of food sites, and the paths to access them vary as well [1]. Dietary shifts, or changes in type of sleeping sites, are examples of primate's reactive responses to habitat alterations [2]. To adapt, non-human primates must inhibit their routine in favour of new behaviours, and this process involves cognitive flexibility. Cognitive flexibility is an executive function [3] which is defined as the ability to switch between different sets of responses. This process is a multi-dimensional one. It involves attentional, memory and learning processes to disengage from the previous strategy and learn a new one [4].

Cognitive flexibility is traditionally tested in humans with the Wisconsin Card Sorting Test (WCST; [5]). During WCST testing, the subject must sort cards considering one dimension

**Competing interests:** The authors have declared that no competing interests exist.

among three possible ones (i.e., number, colour and shape). The dimension to retain (e.g., the colour) is relevant during a certain number of trials, then the rule changes and another dimension (e.g. the number) must be used to sort the cards in the next trials. Cognitive flexibility is evaluated by the number of perseverative errors (i.e., responses based on the previous rule) made by the subject after the rule change.

In non-human primates, cognitive flexibility has been frequently investigated with the Conceptual Set-Shifting Task (CSST, e.g., [6, 7]), which is an adaptation of WCST. In the CSST, monkeys are trained to identify a target from a set of distractors using a given rule (e.g. touch the circle shape independently of its colour) until the relevant dimension changes (e.g., touch the red object independently of its shape). As in the WCST, cognitive flexibility is inferred from the number of perseverative errors following the rule change. In our laboratory, the use of the CSST in Guinea baboons (*Papio papio*) has previously demonstrated an effect of age on cognitive flexibility [8, 9]. Cognitive flexibility was improved from the juvenile age to adulthood. By contrast, cognitive flexibility showed an initial impairment in middle-aged individuals, and was drastically impaired in the oldest baboons. Similar effects of aging on cognitive flexibility were reported in mouse lemurs (*Microcebus murinus*) [10], rhesus macaques (*Macaca mulatta*) [6, 7, 11], chimpanzees (*Pan troglodytes*) [12], and humans [13–16], suggesting that this aging profile is shared across primates.

If cognitive flexibility helps primates adapt to contextual changes in general, then this function should also serve to adapt to social changes [17]. Several studies support the hypothesis that a relation exists between cognitive flexibility and adaptation to social changes in non-human primates. For instance, Amici and collaborators [18, see 19 for an erratum] compared cognitive flexibility in primates with a complex fission-fusion social structures requiring frequent adaptations to social changes, to species with more cohesive, less-changing, social structures. They found that primates with a fission-fusion social structure (i.e., chimpanzees (*Pan troglodytes*), bonobos (*Pan paniscus*), orangutans (*Pongo pygmaeus*) and spider monkeys (*Ateles geoffroyi*)) performed better in five different tasks involving inhibitory control and cognitive flexibility than the more cohesive species (i.e, brown capuchins (*Cebus apella*), long-tailed macaques (*Macaca fascicularis*) and gorillas (*Gorilla gorilla*)). From a different perspective, Izquierdo and collaborators [20] compared performance in cognitive flexibility in rhesus macaques carrying several variants of the gene encoding the serotonin transporter (5-HTT). Macaques carrying both copies of the short allele (SS) of the gene 5-HTTLPR showed a significant impairment in flexibility in conjunction with an alteration of their socioemotional behaviours (more aggression during exposure to a fake snake and human intruder), in comparison to macaques homozygous for the long allele (LL) and heterozygous for both versions (LS). The above studies therefore suggest that cognitive flexibility serves the adaptation to contextual changes in the social as well as in the non-social domains.

The main goal of the current research was to further explore the contribution of cognitive flexibility to social behaviour in baboons. Amici et al. [18] have demonstrated that primate species with different social structures differ in cognitive flexibility. Our goal here was to verify if a relation also exists between cognitive flexibility performance at the level of the individuals and the position of these individuals within their social. Our study reanalysed in a novel way a unique data set that has been peer-reviewed and published in Gullstrand et al. [8]. This study, which involved 18 Guinea baboons tested over two years with a computerized adaptation of the CSST task, mostly focused on the effect of age on cognitive flexibility. In the current research, we reanalysed the same dataset to further investigate if the dominance rank and social network centrality of the individuals within their social groups at the time of testing can also account for their level of cognitive flexibility. To our knowledge, this study is the first to demonstrate a link between performance in cognitive flexibility and indices of sociality within

a group of non-human primates. It is also an important new contribution to research into inter-individual variability in executive functions.

## Methods

### Ethical statements

This research adhered to French and E.U regulation for the ethical treatment of research animals. It received ethical approval from the National French ethics committee « Comité d'Ethique CE-14 » for experimental animal research, as well as the French Ministry of Education (approval APAFIS#2717–2015111708173794 10 v3). The research protocol minimized potential suffering, because the baboons participated voluntarily to the task and could stop contributing at will. This study did not involve food deprivation, capture, anaesthesia or animal sacrifice at the end of the study.

### Animals and living conditions

The participants are 18 Guinea baboons (*Papio papio*, 6 males and 12 females) living together as a social group at the CNRS Primates Center in Rousset (France). The group of baboons is housed in a 25 X 30 m outdoor enclosure (Fig 1A and 1B) which is enriched with stones and tree trunks for climbing and connected by tunnels to indoor housing used at night. The baboons are not water or food deprived. Water is provided *ad libitum* in the enclosure. Feeding, which includes fruits, vegetables and monkey chow, is delivered daily at 4 PM. Animal health and well-being are monitored daily by trained keepers, veterinarians, anred an ethologist.

Table 1 reports the sex and mean age of the participants during the study period. Age classes were defined as follows. "Young": up to 60 months (5 years old max); "Adult": from 61–130 months (5–10.8 years old); "Middle-age": from 131 to 200 months (10.8–16.7 years old), and "Old": from 201 to 300 months (16.7 years old and more). The "Young" class corresponds to the juvenile period until puberty. The "Adult" class includes sexually mature individuals with the development of secondary sexual characteristics, ready for reproduction, which corresponds to young adults. The "Middle-age" class corresponds to older adults, and the "Old-class" corresponds to the period of life approaching and exceeding the life expectancy in the natural environment.

### Apparatus and social context

The baboons remain in their social group for testing and have a free access from their enclosure to 10 operant conditioning test systems, referred to as Automated Learning Device for Monkeys (ALDM, see [21], see Fig 1). The testing booths are in two trailers installed along the enclosure and were directly accessible through openings made in the wire mesh of their enclosure (see Fig 1A). At any time, baboons can quit the enclosure to enter one of the 10 ALDM systems for testing. Each ALDM (see Fig 1C) comprises a RFID equipment allowing the identification of the subject within the test booth, as well as a touch screen for stimulus presentation and a food dispenser (see [21] for technical details). The baboons are presented with a cognitive task (see below) once identified by an ALDM test system.

### Computerized task and database to infer cognitive flexibility

This research re-used a cognitive database already published in Gullstrand et al. [8], which resulted from the presentation of an adaptation of the WCST to baboons. In brief, in each trial three stimuli appeared in a randomized position on the screen, and the baboons had to select

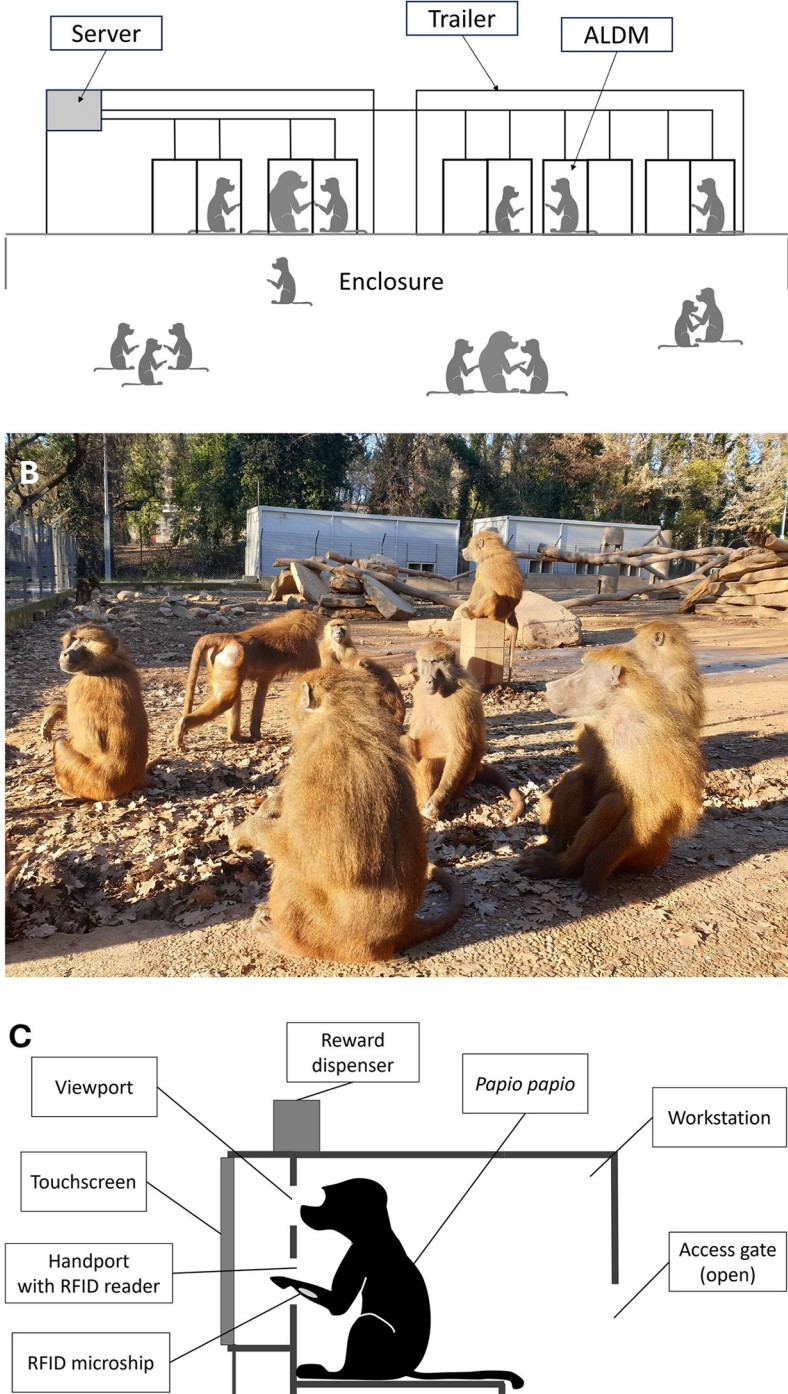

**Fig 1. Research facility.** (A) View of the trailers containing the ALDM test units. These units were freely accessible from the baboon's enclosure; (B) Picture of the enclosure; (C) Schematic view of an ALDM test unit.

**Table 1. Sex, age in months, age-class, and number of rule sessions (RS) used in data analysis after filtering.**

| Name | Sex | MeanAge | Age Class | RS |
|---|---|---|---|---|
| LIPS | F | 49 | Young | 987 |
| LOME | M | 54 | Young | 1324 |
| MALI | F | 48 | Young | 869 |
| NEKKE | F | 31 | Young | 85 |
| EWINE | F | 124 | Adult | 1576 |
| FANA | F | 117 | Adult | 1013 |
| FELIPE | M | 114 | Adult | 218 |
| FEYA | F | 111 | Adult | 1108 |
| HARLEM | M | 88 | Adult | 647 |
| ANGELE | F | 175 | Middle-age | 313 |
| ARIELLE | F | 170 | Middle-age | 1399 |
| BOBO | M | 165 | Middle-age | 63 |
| VIOLETTE | F | 180 | Middle-age | 1672 |
| ATMOSPHERE | F | 263 | Old | 438 |
| BRIGITTE | F | 270 | Old | 39 |
| KALI | F | 292 | Old | 48 |
| PETOULETTE | F | 249 | Old | 37 |
| PIPO | M | 249 | Old | 55 |

and touch the target stimulus considering a rule (see Fig 2). The three stimuli shown in each trial resulted from the combination of three shapes (triangle, circle, "splash") and three colours (yellow, pink, green). Testing was organised in Rule sessions corresponding to one or several blocks of 100 trials in which the same rule applied. Within a rule session, the rule was to either select the stimulus of a given colour (e.g., green) independently of its shape, or to select a

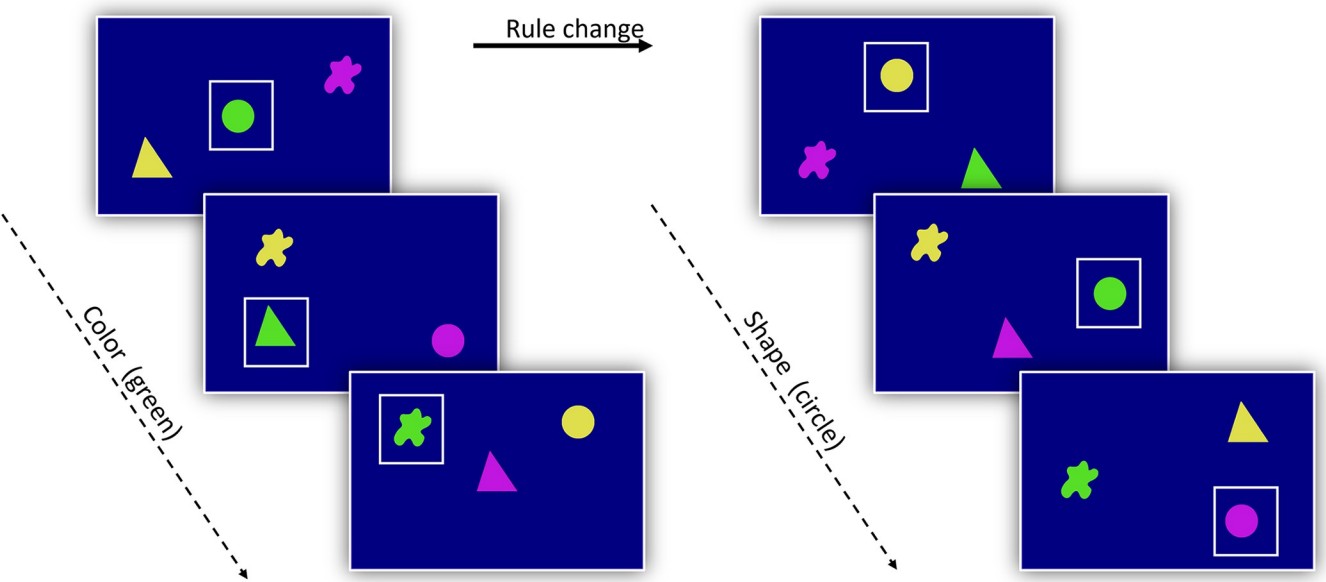

**Fig 2. Illustration of our adaptation of the Wisconsin Card Sorting Test.** On the left, the rule is to find the green colour target. If the success criteria is reached (> = 80% success in a 100-trials block), the rule changes, as shown on the right where the target is now the circle shape. Note that the white frame indicating the positive stimulus in the figure did not appear on real experimental displays.

stimulus of a given shape (e.g., circle) independently of its colour. Baboons were systematically rewarded when correct. Incorrect responses gave rise to a 3 second time-out signalled by a green screen display. The average score of the individuals was automatically calculated at the end of each 100-trial block. If the score remained below 80%, then the baboon resumed an additional 100-trial block conserving the same rule as in the preceding block. If the monkey reached 80% correct or more, then it was now exposed to a new block of 100 trials with a new rule. Monkeys could leave the ALDMs at any time to resume their other activities in their social group.

Data were collected from May 2018 to December 2020, during which the WCST was presented to baboons whenever they had finished other intervening research programs. Use of WCST as a filler task induced many interruptions within the test blocks which imposed a stringent filtering of the data set. In short, we discarded all incomplete rule sessions, as well as the rule sessions interrupted for more than three days. We also applied additional filtering which are described in detail in Gullstrand et al. [8]. Overall, we analysed an average of 577 rule sessions per participant, and a total 14997 set shifting at the group level.

We will retain below three main dependant variables from this data set. They include (1) the frequency of perseverative errors, corresponding to number of responses in accordance with target rule of the previous rule session after a rule change, (2) the learning latency corresponding to the mean number of trials necessary to reach a criterion of ten consecutive successful trials following a rule change; and (3) the mean response time in ms. Perseveration and mean response times were computed from the 50 trials following a rule change.

## Computing social attributes

To achieve our research goal, we needed an accurate picture of the social network of the baboons and their dominance hierarchy when they were tested with the WCST task. To do so, we used two methods already published by our research group in [22] and [23]. In the first study [22], we compared the social network of the baboons inferred from affiliative behaviour observed within the enclosure (i.e., the most commonly used method to compute social networks) to the proximity network inferred from the pairs of individuals using simultaneously adjacent ALDM systems. This analysis revealed a significant and positive correlation between the two networks through different years ($r = 0.7$), suggesting that the proximity network obtained from ALDM testing is a reliable picture of the social network in baboons. In the second study [23], we compared the dominance hierarchy inferred from agonistic behaviours within the enclosure (i.e., which is the most traditional method to compute social ranking) to the dominance hierarchy obtained from the analysis of supplanting behaviours within the ALDM. Supplanting behaviors were defined as events when a baboon takes the place of a social partner when it enters an ALDM system. Again, we found a significant and positive correlation ($r = 0.9$) between the two types of dominance hierarchies through different years, suggesting that the supplants within the ALDM provide an accurate picture of social ranking.

Based on the above two studies, the current research used the proximity network and supplants within the ALDM to infer the social network and social ranking of our baboons, respectively. In practice, the social activity within the ALDM systems were recorded between May 2018 and June 2020. During that period, we used as main computer task either solo tasks (including the WCST) or tasks imposing forms of social coordination between individuals (e.g., [24]). To guarantee that our social measurements were accurate and reliable, we excluded from analyses the time slots during which coordination tasks were used and retained solo tasks periods only. The retained test periods were from June 1st to September 1st, 2018 (Phase 1), from March 1st to June 11th, 2019 (Phase 2) and from September 18th to December 24th, 2019

**Table 2. Social data for each participant.**

| Name | Elo scores | Dominance rank | Eigenvector Centrality |
|---|---|---|---|
| ANGELE | 1114,8 | 2 | 0,219 |
| ARIELLE | 812,5 | 17 | 0,25 |
| ATMOSPHERE | 749 | 19 | 0,088 |
| BOBO | 1034,5 | 6 | 0,041 |
| BRIGITTE | 709,1 | 20 | 0,011 |
| EWINE | 934,1 | 13 | 0,312 |
| FANA | 859,9 | 15 | 0,264 |
| FELIPE | 1215,5 | 1 | 0,179 |
| FEYA | 799,2 | 18 | 0,255 |
| HARLEM | 1061,2 | 5 | 0,191 |
| KALI | 846,5 | 16 | 0,067 |
| LIPS | 912,5 | 14 | 0,306 |
| LOME | 1062,3 | 4 | 0,253 |
| MAKO | 1032 | 8 | 0,3 |
| MALI | 978,2 | 11 | 0,264 |
| MUSE | 936,3 | 12 | 0,317 |
| NEKKE | 981,1 | 10 | 0,172 |
| PETOULETTE | 1032,8 | 7 | 0,073 |
| PIPO | 982,6 | 9 | 0,038 |
| VIOLETTE | 1076,6 | 3 | 0,346 |

(Phase 3). The number of trials collected within each phase were equal to 983 357, 1 926 118, and 1 514 694 trials during phases 1 to 3, respectively.

For each test phase, we assessed copresence network of the baboons in ALDM systems and calculated the Eigenvector centrality (EvC) for each baboon in this network using the method described in [22]. Individual EvCs reflect the tendency of each baboon to be well connected to other members of their group (an EvC close of 0 indicates a lack of connection). We also inferred hierarchical ranks for each phase with the methods described in [23]. For statistical analyses, supplanting behaviors were noted in a winner/loser dataset, and then analyzed with an Elo-rating script which assigned each monkey an Elo score indicating its social ranking (highest score for the most dominant, lowest for the most subordinate).

To ensure the homogeneity of the social data, we computed the Spearman correlation over the three phases, independently for each dependent social variable. In the case of EvC, the correlations were equal to 0.70, 0.86 and 0.73 between Phases 1–2, 1–3 and 2–3 respectively. For the ranks, these correlations were of 0.85, 0.90 and 0.97, respectively. We concluded from these analyses that the social data were homogeneous for the three test phases (see S1 Table), and thus grouped the three phases to obtain a unique EvC index, and a unique Elo-score, covering the two years of testing. The social indexes resulting from this grouping are summarized in Table 2.

## Relating cognitive performance to social attributes

The main aim of this study was to investigate the relationship between the two social indexes (EvC and dominance ranks), and the three measures of cognitive flexibility (perseveration, learning latency and RT). For this purpose, we used mixed-effects models (LMER; lmertest package on R, version 4.2.2), considering random effects to account for repeated measurement (with intercept and slope by subjects) and fixed effects of rank, EvC and age-class (remember

that age was found to be an important factor in Gullstrand et al. [8]). All the models can be summarised by the present formula:

**cognitive flexibility performance ~ Rank + EvC + Age class + (Rule Sessions) | Name)**
Statistical power was computed using Simr package in R [25].

## Results

The detailed results of each model will be described in the following subsections.

### LMER on perseveration

Confirming the results already obtained in [8], the first model shows a strong effect of age-class on perseveration (see Table 3). According to the results of this model, adults perform better than other age groups (p = 0.0015): Baboons in this age class made significantly fewer perseverative errors (mean = 3.34; SE = 0.36) than youngsters (mean = 8.25; SE = 2.02), middle-aged subjects (mean = 8.24; SE = 3.3) and older subjects (mean = 12.68; SE = 2.05). The results of this model show no significant effect (p = 0.18; statistical power [95% CI] = 42.60% [39.51, 45.73]) of the rank on perseveration but there was a highly significant effect (p = 0.005) of centrality on perseveration, the most central individuals made less perseverative errors (Fig 3).

**Formula: Mean Perseveration ~ Rank + EvC + Age class + (scale (Rule Sessions) | Name)**

### LMER on learning latencies

For the learning latencies (see Table 4), we also find an effect of age class. The adults learned significatively faster (p = 0.006) than the other age classes. Adult baboons reached the criteria of ten consecutive successful trials with a smaller number of trials (adults mean = 21.9; SE = 1.35), than the young baboons (young mean = 39.6; SE = 8.75), the middle age (mean = 37.8; SE = 10.5) and the oldest individuals (mean = 57.0; SE = 9.32). As for the perseverative errors, the analysis revealed no significant effect of rank on learning latencies (p = 0.77; statistical power [95% CI] = 16.80% [14.53, 19.26]). On the contrary, the analyses revealed a significant effect of centrality on learning latencies (p = 0.012) corresponding to a better performance in the most central individuals (see Fig 3).

**Table 3. Results of the linear mixed effect model on the average of perseverative errors on the first 50 trials after a rule change.**

| Random effects: | | | | | | |
|---|---|---|---|---|---|---|
| Groups | Name | Variance | SD | Corr | | |
| Name | (Intercept) | 47.06 | 6.860 | | | |
| | Scale (Rule Sessions) | 0.14.63 | 0.3.825 | 0.93 | | |
| Residual | | 12.12.47 | 3.531 | | | |
| Fixed effects: | | | | | | |
| | Estimate | SE | df | t value | p-value | |
| **(Intercept)** | **22.066** | **2.987** | **13.860** | **7.388** | **3.62e-06** | *** |
| Rank | -0.165 | 0.117 | 11.559 | -1.410 | 0.18484 | |
| **EvC_All_Parc** | **-35.140** | **10.785** | **14.966** | **-3.258** | **0.00531** | ** |
| **AgeClassAdult** | **-7.652** | **1.767** | **11.181** | **-4.331** | **0.00115** | ** |
| AgeClassMiddleAge | -3.700 | 1.922 | 11.477 | -1.925 | 0.07941 | . |
| AgeClassOld | -5.020 | 2.884 | 13.569 | -1.741 | 0.10436 | |

Note. Number of observations: 4312, number of individuals: 18

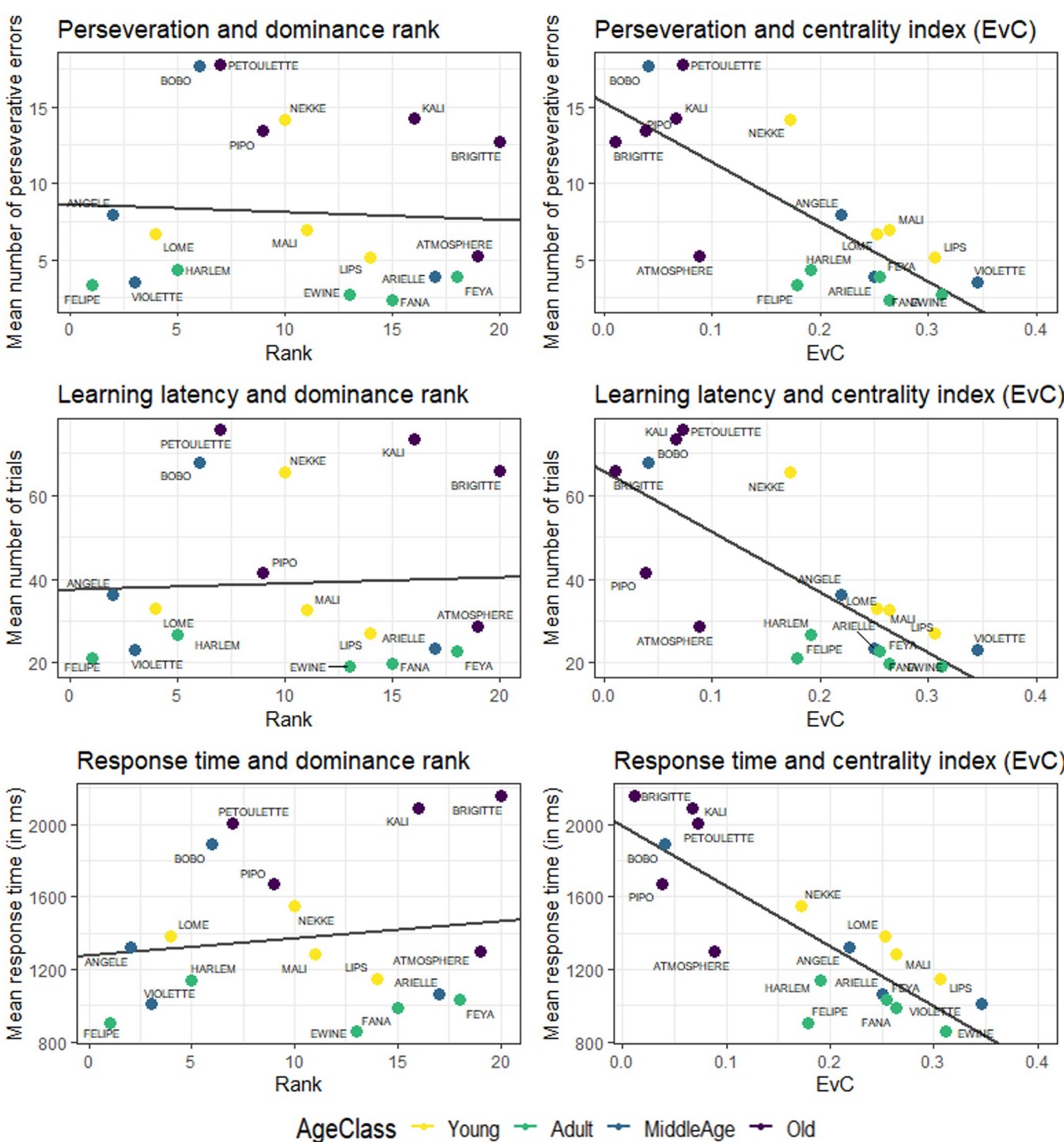

**Fig 3. Cognitive flexibility performance as a function of hierarchical rank and centrality index.** The colours indicate the different age classes, and the line corresponds to the linear regression between the considered dependent variable and either the hierarchical rank (left panel) or the centrality index of each subject (EvC; right panel).

**Table 4. Results of the linear mixed effect model on the learning latency after a rule shift.**

| Random effects: | | | | | | |
|---|---|---|---|---|---|---|
| Groups | Name | Variance | SD | Corr | | |
| Name | (Intercept) | 107296.5 | 327.56 | | | |
| | Scale (Rule Sessions) | 42654.9 | 206.53 | 1 | | |
| Residual | | 211.9 | 14.56 | | | |
| Fixed effects: | | | | | | |
| | Estimate | SE | df | t-value | p-value | |
| **(Intercept)** | **91.7588** | **14.4192** | **12.6987** | **6.364** | **2.77e-05** | *** |
| Rank | -0.1647 | 0.5693 | 10.8048 | -0.289 | 0.77777 | |
| **EvC** | **-150.8149** | **52.0234** | **13.7207** | **-2.899** | **0.01187** | * |
| **Age Class: Adult** | **-29.6062** | **8.5753** | **10.3638** | **-3.452** | **0.00589** | ** |
| Age Class: MiddleAge | -17.1269 | 9.3316 | 10.7284 | -1.835 | 0.09430 | . |
| Age Class: Old | -23.0247 | 13.9485 | 12.4821 | -1.651 | 0.12372 | |

Note. Number of observations: 4312, number of individuals: 18

**Formula: Learning latency ~ Rank + EvC + Age class + (scale (Rule Sessions) | Name)**

## LMER on response times

Response time analyses also revealed an effect of age-class (p = 0.006, see Table 5). The response times of adults were significantly shorter (mean = 984ms; SE = 50) than those of the young (mean = 1340ms; SE = 84), middle-aged (mean = 1320ms; SE = 201) and older baboons (mean = 1844ms; SE = 160). In addition, there were no significant effect of the rank on response times (p = 0.73; statistical power [95% CI] = 16.50% [14.25, 18.95]). By contrast, an effect of centrality emerged, corresponding to significantly shorter response times for the most central individuals (p = 0.024; see Fig 3).

**Formula: Mean RT ~ Rank + EvC + Age class + (scale (Rule Sessions) | Name)**

## Correlation tests between social indexes and flexibility performances

Spearman correlations between the dominance ranks and performance were not significant (see Fig 3). In the case of perseverative errors, this correlation was rs = -0.03 (p-value = 0.9; CI

**Table 5. Results of the linear mixed effect model on the average RT on the first 50 trials after a rule shift.**

| Random effects: | | | | | | |
|---|---|---|---|---|---|---|
| Groups | Name | Variance | SD | Corr | | |
| Name | (Intercept) | 105834 | 325.3 | | | |
| | Scale (Rule Sessions) | 24176 | 155.5 | 0.75 | | |
| Residual | | 47375 | 217.7 | | | |
| Fixed effects: | | | | | | |
| | Estimate | SE | df | t-value | p-value | |
| **(Intercept)** | **2105.974** | **239.459** | **13.209** | **8.419** | **1.14e-06** | *** |
| Rank | 2.399 | 9.577 | 11.749 | 0.251 | 0.80651 | |
| **EvC_All_Parc** | **-2171.140** | **858.027** | **13.967** | **-2.530** | **0.02405** | * |
| **AgeClassAdult** | **-483.706** | **145.135** | **11.544** | **-3.333** | **0.00627** | ** |
| AgeClassMiddleAge | -187.655 | 157.445 | 11.796 | -1.192 | 0.25674 | |
| AgeClassOld | -81.492 | 231.830 | 12.988 | -0.352 | 0.73084 | |

Note. Number of observations: 4312, number of individuals: 18

95% = [-0.47; 0.46]). This correlation was of rs = -0.007 (p-value = 1; CI 95% = [-0.47; 0.48]) for the learning latencies and rs = 0.13 (p-value = 0.6; CI 95% = [-0.39; 0.64]) for the response times. By contrast, the Spearman correlations between performance and the EvC centrality indexes were all significant (see Fig 3): These correlations were equal to -0.73 (CI = [-0.88; -0.46]; p-value = 0.0006), -0.74 (CI = [-0.87; -0.46]; p-value = 0.0004), and -0.78 (CI = [-0.92; -0.39]; p-value = 0.0001), respectively for the perseverations, learning latencies and response times.

## Discussion

Below, we will sequentially discuss the effects of age, rank, centrality, and the relationships between these factors.

### Age effects

Our findings confirm that cognitive flexibility is age-dependent in baboons. Adult baboons in our study expressed a lower perseveration rate, faster learning latencies, and faster response times than the other three age groups, suggesting that they outperformed their social partners in terms of cognitive flexibility. The excellent performance of the adult age-group converges with Gullstrand et al. [8] conclusions' that cognitive flexibility remains in a developmental stage until adulthood and starts to decline in efficiency in the middle-aged group and even further in the oldest age-group. Age effects in our study are also congruent with previous results obtained on humans [13–16] and non-human primates (in chimpanzees [12]; rhesus macaques [6, 7, 11]; and mouse lemur [10]).

### Hierarchical rank

Guinea baboons display a high degree of social tolerance [26, 27], which makes it difficult to assess dominant hierarchies (see [26, 28]). In our study we used supplanting behaviours during cognitive testing to alleviate this difficulty. We could thus address the critical question of whether cognitive flexibility is related to hierarchical ranks, and we found no systematic relationship between these factors, independently of the dependent variables considered (perseveration, learning latency, or response time).

Two reasons lead us to believe that a sample size of 18 subjects might not be sufficient to reveal a potential relationship between rank and cognitive flexibility.

First, cognitive flexibility varies greatly with age and/or training in Guinea baboons [1], in contrast to ranks which are much more stable during lifetime (personal observations). Given the relative stability of ranks, identifying the potential link between rank and cognitive flexibility might require sample sizes much larger than ours. In our study, the statistical power of our analyses remained well below 80% for ranks, suggesting that we had too few subjects to detect the subtlety of the relationship between flexibility and rank.

Second, Johnson-Ulrich & Holekamp [29] recently studied cognitive flexibility in spotted hyenas (*Crocuta crocuta*), using a task requiring to bypass a barrier. This study showed an effect of rank on cognitive flexibility only when the group size reaches a critical threshold, with lower-ranked hyenas showing a better inhibitory control than higher-ranked hyenas. According to the authors, this effect can be explained by an increase social competition for resources when the group size increases. A similar effect might emerge in baboons if social pressure increases with group size, supporting the view that hierarchical rank only becomes important when resource competition increases.

## Centrality

Amici et al. [18, 30] found that non-human primates with fission-fusion have a more efficient inhibition and cognitive flexibility, compared to other species with a more cohesive social structure, suggesting that executive functions and social behaviours are not independent factors. Here we found a statistically significant correlation between cognitive flexibility and social network centrality, for our three dependent variables (perseveration, learning latency and response time), the most central baboons demonstrated an enhanced cognitive flexibility in comparison to more peripheral ones. This result adds to Amici et al.'s [18] findings because our results were obtained from the comparisons of individuals belonging to the same social group and species, and not from inter-species comparisons. Our results also converge with those of Bonino & Cattelino [31] in humans, indicating that the most cooperative children, and those able to solve social conflicts, were precisely those with the best performance in cognitive flexibility. In summary, results obtained on humans, baboons, and from the comparison of different primate species all suggest that cognitive flexibility is a crucial element in social problem-solving.

## On the relationship between cognitive flexibility and sociality

The fact that centrality correlates with cognitive flexibility can be explained in two different ways. The first possibility is that cognitive flexibility helps individuals navigate different strategies to respond to social problems appropriately. The ability to choose the best strategy for solving social problems would enhance the individual's social status, giving it a more central position in the social network. Conversely, an individual acting spontaneously without inhibiting inappropriate strategies, and/or acting rigidly in different contexts, would find itself in a more unfavourable social position, at the periphery of the social network.

The second possibility is that sociality is the leading factor. Considering that cognitive flexibility improves with practice [8], it can be proposed that the most central individuals have an enhanced cognitive flexibility because they are exposed to more frequent, and more complex, social problems than the more peripheral baboons of the network. This conclusion is in agreement with [29], that hyenas (*Crocuta crocuta*) living in larger groups showed better performance in terms of cognitive flexibility.

The above two hypotheses are not mutually exclusive. It is indeed also possible that sociality and cognitive flexibility influence each other, and that a positive feedback loop drives the evolution of both. This prediction agrees with the social brain hypothesis [32], suggesting that the complexity of sociality shaped brain development during primate evolution. Additional studies are needed to better understand the nature and extent of the relationships between cognitive flexibility and sociality.

## Centrality and age

Our statistical analyses showed that efficiency of cognitive flexibility depends on the age of the individuals, with the best performance being expressed in adults. They further revealed that cognitive flexibility and centrality are two correlated factors. These two results question the relationship between age and the centrality of individuals.

We addressed this question by performing various Spearman correlation tests between age and centrality. When all age groups are considered, the Spearman correlation between the age and centrality index is 0.52, which is statistically significant (p = 0.03). As shown in Fig 4 (left panel), centrality decreases with age.

Fig 4 shows that all the oldest subjects (in purple) have a low centrality index (all below 0.1). The other subjects in the younger age groups appear to be more evenly distributed, with

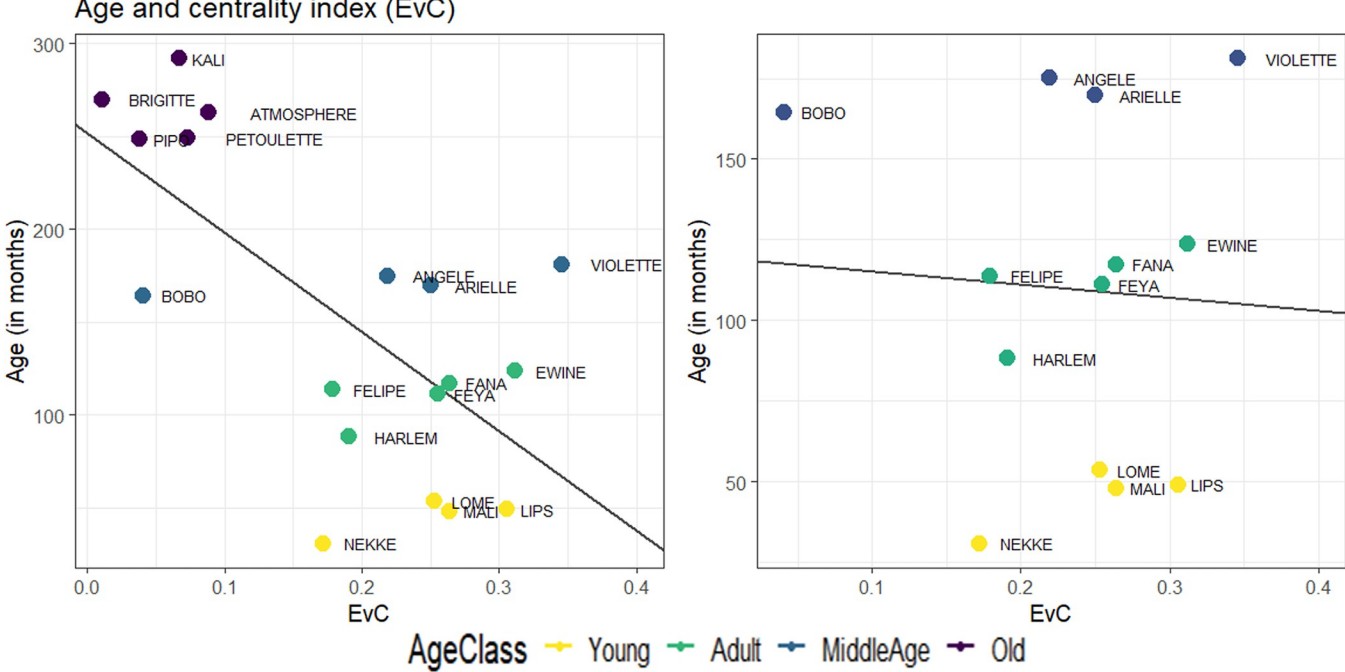

**Fig 4. Correlation between age (in months) and centrality index (EvC).** The left panel illustrates the correlation for all subjects. The right panel illustrates the same correlation with the old class excluded from the dataset. The colours indicate the different age classes and the lines correspond to the linear regression between the age and the centrality index (EvC) of each subject.

different levels of centrality ranging from 0.15 to 3.5. When the youngest age group is discarded from statistical analysis, the Spearman correlation between age and centrality is equal to -0.5, an almost identical correlation but one that becomes non-significant (p = 0.052). Conversely, when the oldest subjects are removed from analysis, the Spearman correlation collapsed to 0.13, with a p-value of 0.68 (see Fig 4, right panel). Thus, the initial correlation of -0.52 computed on all age group is mostly driven by the oldest individuals, who present the lowest centrality. It can therefore be concluded that, beyond a certain age, deficits in cognitive flexibility become so important that they lead to the isolation of individuals. An alternative hypothesis is that the isolation of the oldest individuals exacerbate age-related changes, as observed in humans [33].

Is age the sole driver of the relationship between centrality and cognitive flexibility? If older individuals are both less central and less flexible this may explain the general relationship between centrality and flexibility found previously. Two results allow us to rule out this possibility. Firstly, age and centrality are included in the statistical models, and both are important. Secondly, we repeated the same analyses, without old age class, and found qualitatively similar results (see S2–S4 Tables), although the significance is affected by the disappearance of a quarter of the subjects. The results of these new models show that the effects of age and centrality are robust even without the oldest individuals.

To conclude, the current study confirms that cognitive flexibility and social behaviors are two related factors in baboons. Our study shows that the most central baboons in their social network are precisely those with the best performance in terms of cognitive flexibility. Additional studies are needed to determine if a relation between rank and cognitive flexibility would emerge from the study of larger social groups.

## Supporting information

**S1 Table. Spearman correlation tests (and their confidence interval) on centrality and rank factors, among the 3 phases.**
(DOCX)

**S2 Table. Results of the linear mixed effect model on the average of perseverative errors on the first 50 trials after a rule change.** Note that the "OLD" age class was removed from the dataset in this analysis.
(DOCX)

**S3 Table. Results of the linear mixed effect model on the learning latency after a rule shift.** Note that the "OLD" age class was removed from the dataset in this analysis.
(DOCX)

**S4 Table. Results of the linear mixed effect model on the average RT on the first 50 trials after a rule.** Note that the "OLD" age class was removed from the dataset in this analysis.
(DOCX)

## Acknowledgments

We thank S. Barniaud and the staff at the Rousset-sur-Arc Primate Center (CNRS-UAR 846, France) for support.

## Author Contributions

**Conceptualization:** Julie Gullstrand, Joël Fagot.

**Formal analysis:** Julie Gullstrand, Nicolas Claidière.

**Funding acquisition:** Joël Fagot.

**Methodology:** Julie Gullstrand, Nicolas Claidière, Joël Fagot.

**Resources:** Nicolas Claidière.

**Software:** Nicolas Claidière, Joël Fagot.

**Supervision:** Joël Fagot.

**Writing – original draft:** Julie Gullstrand, Nicolas Claidière, Joël Fagot.

**Writing – review & editing:** Julie Gullstrand, Nicolas Claidière, Joël Fagot.

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
