## [Decision Letter · Decision Letter 0]

9 Sep 2024

PONE-D-24-31940Cognitive flexibility and sociality in Guinea baboons (Papio papio)PLOS ONE

Dear Dr. Gullstrand,

Thank you for submitting your manuscript to PLOS ONE. After careful consideration, we feel that it has merit but does not fully meet PLOS ONE’s publication criteria as it currently stands. Therefore, we invite you to submit a revised version of the manuscript that addresses the points raised during the review process.

Both reviewers see value in this manuscript, as do I. Reviewer 1 has only minor comments, whereas Reviewer 2 requests more clarity and details in some sections and more depth in the interpretations. I’m happy to invite a revised manuscript that attends to such issues. In that revision, please include a power analysis for the non-significant findings – I doubt that statistical power was the reason for the null results, but it would strengthen the paper to show that the null results weren’t just caused by low statistical power. Ideally, you would do an equivalence test or Bayesian analysis for the non-significant findings, to show that the non-significant findings really are null effects, but that’s going beyond the current norms and expectations for most papers, so a power analysis will suffice. I look forward to reading your revised manuscript.

We look forward to receiving your revised manuscript.

Kind regards,

Pat Barclay

Academic Editor

PLOS ONE

Journal requirements: 1. When submitting your revision, we need you to address these additional requirements. Please ensure that your manuscript meets PLOS ONE's style requirements, including those for file naming. The PLOS ONE style templates can be found at https://journals.plos.org/plosone/s/file?id=wjVg/PLOSOne_formatting_sample_main_body.pdf and https://journals.plos.org/plosone/s/file?id=ba62/PLOSOne_formatting_sample_title_authors_affiliations.pdf. 2. In order to comply with PLOS ONE's guidelines for non-human primate experiments (http://journals.plos.org/plosone/s/submission-guidelines#loc-non-human-primates), please provide additional details regarding housing conditions, feeding regimens, environmental enrichment. Also indicate how often animal care staff monitored the health and well-being of the animals and the criteria used to make such assessments. Lastly, specify the disposition of animals at the end of the study (e.g. returned to home colony, euthanasia, etc.). If any animals were euthanized following the study, please provide the method of sacrifice. 3. Please note that PLOS ONE has specific guidelines on code sharing for submissions in which author-generated code underpins the findings in the manuscript. In these cases, all author-generated code must be made available without restrictions upon publication of the work. Please review our guidelines at https://journals.plos.org/plosone/s/materials-and-software-sharing#loc-sharing-code and ensure that your code is shared in a way that follows best practice and facilitates reproducibility and reuse. 4. We noted in your submission details that a portion of your manuscript may have been presented or published elsewhere. [Our study re-used the same data set as in Gullstrand et al,. Behavioural Brain Research 2022;114043, but provided a new look at these data. Here we examined the relationship between social factors (dominance rank and centrality) and cognitive flexibility. The previous paper only considered age effects on flexibility.] Please clarify whether this publication was peer-reviewed and formally published. If this work was previously peer-reviewed and published, in the cover letter please provide the reason that this work does not constitute dual publication and should be included in the current manuscript. 5. When completing the data availability statement of the submission form, you indicated that you will make your data available on acceptance. We strongly recommend all authors decide on a data sharing plan before acceptance, as the process can be lengthy and hold up publication timelines. Please note that, though access restrictions are acceptable now, your entire data will need to be made freely accessible if your manuscript is accepted for publication. This policy applies to all data except where public deposition would breach compliance with the protocol approved by your research ethics board. If you are unable to adhere to our open data policy, please kindly revise your statement to explain your reasoning and we will seek the editor's input on an exemption. Please be assured that, once you have provided your new statement, the assessment of your exemption will not hold up the peer review process. 6. Your abstract cannot contain citations. Please only include citations in the body text of the manuscript, and ensure that they remain in ascending numerical order on first mention. 7. Please include captions for your Supporting Information files at the end of your manuscript, and update any in-text citations to match accordingly. Please see our Supporting Information guidelines for more information: http://journals.plos.org/plosone/s/supporting-information.

Reviewers' comments:

Reviewer's Responses to Questions

**Comments to the Author**

1. Is the manuscript technically sound, and do the data support the conclusions?

Reviewer #1: Yes

Reviewer #2: Yes

2. Has the statistical analysis been performed appropriately and rigorously? 

Reviewer #1: Yes

Reviewer #2: Yes

3. Have the authors made all data underlying the findings in their manuscript fully available?

Reviewer #1: Yes

Reviewer #2: Yes

4. Is the manuscript presented in an intelligible fashion and written in standard English?

Reviewer #1: Yes

Reviewer #2: Yes

5. Review Comments to the Author

Reviewer #1: This paper examined the relationship between cognitive flexibility and social structure. It was found that social rank did not show any significant relationship with measures of cognitive flexibility, but that there was a significant relationship with centrality within the social network. These findings were obtained for data that was re-analyzed from a study on 18 Guinea baboons in captivity that were trained on a free access computer system to perform a Conceptual Set-Shifting Task (CSST). I found the study to be straightforward, appropriate for testing the hypothesis, and with results that make sense and improve our understanding of cognition and social behavior.

The authors have shown us how cognitive flexibility, which is the ability to switch between different task fluidly, can relate to social demands. Cognitive flexibility involves attention, memory, learning, and the ability to disengage from one task and focus on learning another. It is argued that more complex social systems demand greater cognitive flexibility, citing that studies have shown individuals from fission fusion societies have better inhibitory control and cognitive flexibility than less complex systems. Furthermore, genotype variation in a serotonin transporter was also related to this.

The authors set out to test this idea and were successful at showing that individuals with the greatest social demands (i.e., the most central individuals) tended to show the greatest cognitive flexibility.

Overall, I find everything in good order and I recommend the paper for publication.

Some minor points/questions

In Table 1 – please spell out RS in caption. I had to dig through text after table to understand it was Rule Sessions.

Elo-Scores could be explained better. They are mentioned to be scored from supplants, but please clarify what it is, so that the reader of the paper can understand everything without being referred to another paper.

It seems to me the two proposed possibilities for the result sets up a feedback loop mechanism underlying how more complex sociality and more sophisticated cognition evolve.

Are there any research supporting that the complexity of social interaction relates to any gene expression changes in the brain. Do animal that encounter greater complexity show brain gene expression differences?

Reviewer #2: An interesting article and an innovative study. However, overall the writing is quite informal and lacks depth in the interpretation of the findings. In addition, the computing social attributes sections are very confusing and needs considerably more detail to fully evaluate the task, analysis and interpretation of the data. A few minor concerns:

1. Lin 92 - I think CCST is supposed to be CSST

2. Line 153 - the criterion indicated here does not match the criterion indicated in Line 140.

3. In table 1, what is RS? I am guessing reaction speed, if so, the unit of measure is needed.

4. In table 1, what is the age ration between baboons and humans?

5. Line 330 - I would saying the "efficiency of cognitive flexibility depends on age".

6. PLOS authors have the option to publish the peer review history of their article (what does this mean?). If published, this will include your full peer review and any attached files.

Reviewer #1: No

Reviewer #2: **Yes: **Tara L. Moore

---

## [Author Response · Author response to Decision Letter 0]

21 Oct 2024

Dear Editor, 

We thank you for your suggestions and positive evaluation of our manuscript entitled “Cognitive flexibility and sociality in Guinea baboons (Papio papio)”. We are pleased to submit our revision in two different versions, an unmarked copy and a marked-up copy to track changes.

In the revision, we did our best to more strictly follow PLOS ONE’s recommended style and guidelines. To do so : 

1. We corrected titles and authors affiliations to fit PLOS ONE’s style.

2. We now provide additional details regarding animal suffering in the section on “Ethical statements” (lines 106-109), and have added the requested information in the “Animals and living conditions” section (lines 113-119), regarding animal housing conditions, feeding regimens, environmental enrichment and research protocol.

3. We have made our data sets and codes for statistical analyses available: DOI 10.17605/OSF.IO/JNW82 

4. We made it clear that “Our study reanalysed in a novel way a unique data set that has been peer-reviewed and published in Gullstrand et al. [8]”. The full reference of this paper is “Gullstrand J, Claidière N, Fagot J. Age effect in expert cognitive flexibility in Guinea baboons (Papio papio). Behavioural Brain Research. 2022;114043.”

5. We removed the citation of the above paper in the abstract and we ensured that the citations remain in ascending numerical order on first mention.

6. We now correctly refer to the supporting information file in the manuscript. 

7. We reviewed all the reference list to ensure that it is complete and correct. You raised the issue of potential retracted papers. There is none to our knowledge. The only problematic reference is “Amici F, Aureli F, Call J. Fission-Fusion Dynamics, Behavioral Flexibility, and Inhibitory Control in Primates. Current Biology. 2008;18:1415–9” for which an erratum has been published “Amici F, Aureli F, Call J. Fission-Fusion Dynamics, Behavioral Flexibility, and Inhibitory Control in Primates. Current Biology. 2013;23:1267”. We now cite this erratum which does not alter the general conclusion of this research.

You letter also raised the issue of statistical power. To address this concern, we followed your suggestion to add power analyses in our manuscript, to better interpret the lack of relationships between rank and cognitive flexibility. Power analyses were thus made a posteriori with Simr package in R on the effects of rank. The results are reported lines 229 257 and 270. Our original version of our manuscript already mentioned the possibility that we had too few subjects to detect an effect of cognitive flexibility on ranks. The statistical results (statistical power was systematically below 50% for ranks) reinforce this conclusion. We have now added a section in the discussion to address that point (lines 311 – 319). Note that the statistical power was not a concern for the other discoveries of our research, especially for our main significant finding that the most central baboons were those showing the greatest cognitive flexibility.

We hope that you will be satisfied with this revision and that you will consider that this manuscript is now ready for publication.

Comments to reviewer #1

Thank you for a positive and constructive evaluation of our research. You will find below a copy of your comments and our response to them.

This paper examined the relationship between cognitive flexibility and social structure. It was found that social rank did not show any significant relationship with measures of cognitive flexibility, but that there was a significant relationship with centrality within the social network. These findings were obtained for data that was re-analyzed from a study on 18 Guinea baboons in captivity that were trained on a free access computer system to perform a Conceptual Set-Shifting Task (CSST). I found the study to be straightforward, appropriate for testing the hypothesis, and with results that make sense and improve our understanding of cognition and social behavior.

The authors have shown us how cognitive flexibility, which is the ability to switch between different task fluidly, can relate to social demands. Cognitive flexibility involves attention, memory, learning, and the ability to disengage from one task and focus on learning another. It is argued that more complex social systems demand greater cognitive flexibility, citing that studies have shown individuals from fission fusion societies have better inhibitory control and cognitive flexibility than less complex systems. Furthermore, genotype variation in a serotonin transporter was also related to this.

The authors set out to test this idea and were successful at showing that individuals with the greatest social demands (i.e., the most central individuals) tended to show the greatest cognitive flexibility.

Overall, I find everything in good order and I recommend the paper for publication.

R#1: Some minor points/questions

In Table 1 – please spell out RS in caption. I had to dig through text after table to understand it was Rule Sessions.

Authors’ Response: We have now replaced RS by “rule session” in the text. However, as the label “rule session” was too long to fit in Table 1, we continued to use RS in the main body of the table but have now defined RS in its legend.

R#1: Elo-Scores could be explained better. They are mentioned to be scored from supplants, but please clarify what it is, so that the reader of the paper can understand everything without being referred to another paper. 

Authors’ Response: We have corrected a section (lines 206-209) to address this comment, and have improved our description of the computation of the social attributes (lines 180-190) . We now explain what is a supplant within the test systems, and provide more details on Elo-scores and their calculation. 

R#1: It seems to me the two proposed possibilities for the result sets up a feedback loop mechanism underlying how more complex sociality and more sophisticated cognition evolve.

Authors’ Response: Thank you for this suggestion. We have now added a full section to mention the possibility of a feedback loop between sociality and cognitive flexibility. Here is what we say (lines 356-361):

“The above two hypotheses are not mutually exclusive. It is indeed also possible that sociality and cognitive flexibility influence each other, and that a positive feedback loop drives the evolution of both. This prediction agrees with the social brain hypothesis [30], suggesting that the complexity of sociality shaped brain development during primate evolution. Additional studies are needed to better understand the nature and extent of the relationships between cognitive flexibility and sociality”

R#1: Are there any research supporting that the complexity of social interaction relates to any gene expression changes in the brain. Do animal that encounter greater complexity show brain gene expression differences?

Authors’ Response: We are not aware of such research, but honestly, this is not our field of expertise.

Comment to reviewer #2 (Dr. Tara L. Moore)

We are very grateful for your comments which have helped us to Improve our manuscript. You will find below a copy of your remarks and a description on how we have addressed them.

Reviewer #2: An interesting article and an innovative study. However, overall the writing is quite informal and lacks depth in the interpretation of the findings. In addition, the computing social attributes sections are very confusing and needs considerably more detail to fully evaluate the task, analysis and interpretation of the data. 

Authors’ Response: To address this concern, we have revised the section on the “Computation of social attributes” to better explain how social networks and dominance hierarchy are recorded and computed (lines 180-194). We also now explain with greater details the rationale for our use of the Elo scores (lines 206-209). At the request of reviewer #1, we have also added an additional hypothesis on the potential relationships between cognitive flexibility and ranks. We hope that these changes will alleviate your concerns. 

R#2: A few minor concerns:

Line 92 - I think CCST is supposed to be CSST

Authors’ Response: Thank you, we have corrected our error.

R#2: Line 153 - the criterion indicated here does not match the criterion indicated in Line 140.

Authors’ Response: These are different values: the first one (now line 154) is the learning criterion to access to a new block of trials with a new rule. The second one (now line 168) is not a criterion. It is the number of trials that we retained after a rule change for the analysis of the learning latencies.

R#2: In table 1, what is RS? I am guessing reaction speed, if so, the unit of measure is needed.

Authors’ Response: RS stands for “rule session”, but we now use the full spelling all along the manuscript. 

R#2: In table 1, what is the age ration between baboons and humans.

Authors’ Response. The life expectancy in captivity in baboons is approximately of 30 years, but we do not think that we should mention this value. To address your point, we have added a section provided the age range for each age-class. This information is given lines 120-128.

R#2: Line 330 - I would saying the "efficiency of cognitive flexibility depends on age".

Authors’ Response: Thank you, we have followed your suggestion.

---

## [Editor Report · Decision Letter 1]

22 Nov 2024

Cognitive flexibility and sociality in Guinea baboons (Papio papio)

PONE-D-24-31940R1

Dear Dr. Gullstrand,

We’re pleased to inform you that your manuscript has been judged scientifically suitable for publication and will be formally accepted for publication once it meets all outstanding technical requirements.

Kind regards,

Pat Barclay

Academic Editor

PLOS ONE

Additional Editor Comments (optional):

The revisions have improved the manuscript, and I am happy to now accept it. There is a minor typo on line 207 ("looser" should be "loser"), but that can be corrected in the proofing stage.
---

## [Editor Report · Acceptance letter]

30 Nov 2024

PONE-D-24-31940R1 

PLOS ONE

Dear Dr. Gullstrand, 

I'm pleased to inform you that your manuscript has been deemed suitable for publication in PLOS ONE. Congratulations! Your manuscript is now being handed over to our production team.

Kind regards, 

on behalf of

Dr. Pat Barclay 

Academic Editor

PLOS ONE